

# A systematic review of the direct and indirect effects of herbivory on plant reproduction mediated by pollination

Stephanie M. Haas[1] and Christopher J. Lortie[1,2]

[1] Department of Biology, York University, Toronto, ON, Canada
[2] The National Center for Ecological Analysis and Synthesis, University of California, Santa Barbara, Santa Barbara, CA, USA

## ABSTRACT

**Background:** Plant reproduction is influenced by the net outcome of plant–herbivore and plant–pollinator interactions. While both herbivore impacts and pollinator impacts on plant reproduction have been widely studied, few studies examine them in concert.

**Methodology:** Here, we review the contemporary literature that examines the net outcomes of herbivory and pollination on plant reproduction and the impacts of herbivores on pollination through damage to shared host plants using systematic review tools. The direct or indirect effects of herbivores on floral tissue and reported mechanisms were compiled including the taxonomic breadth of herbivores, plants and pollinators.

**Results:** A total of 4,304 studies were examined producing 59 relevant studies for synthesis that reported both pollinator and herbivore measures. A total of 49% of studies examined the impact of direct damage to floral tissue through partial florivory while 36% of studies also examined the impact of vegetative damage on pollination through folivory, root herbivory, and stem damage. Only three studies examined the effects of both direct and indirect damage to pollination outcomes within the same study.

**Conclusions:** It is not unreasonable to assume that plants often sustain simultaneous forms of damage to different tissues and that the net effects can be assessed through differences in reproductive output. Further research that controls for other relative drivers of reproductive output but examines more than one pathway of damage simultaneously will inform our understanding of the mechanistic relevance of herbivore impacts on pollination and also highlight interactions between herbivores and pollinators through plants. It is clear that herbivory can impact plant fitness through pollination; however, the relative importance of direct and indirect damage to floral tissue on plant reproduction is still largely unknown.

## INTRODUCTION

Plant fitness is determined in part by the net outcome of interactions with other species. All species within a community experience multiple direct interactions ranging from

Corresponding author
Stephanie M. Haas,
shaas014@yorku.ca

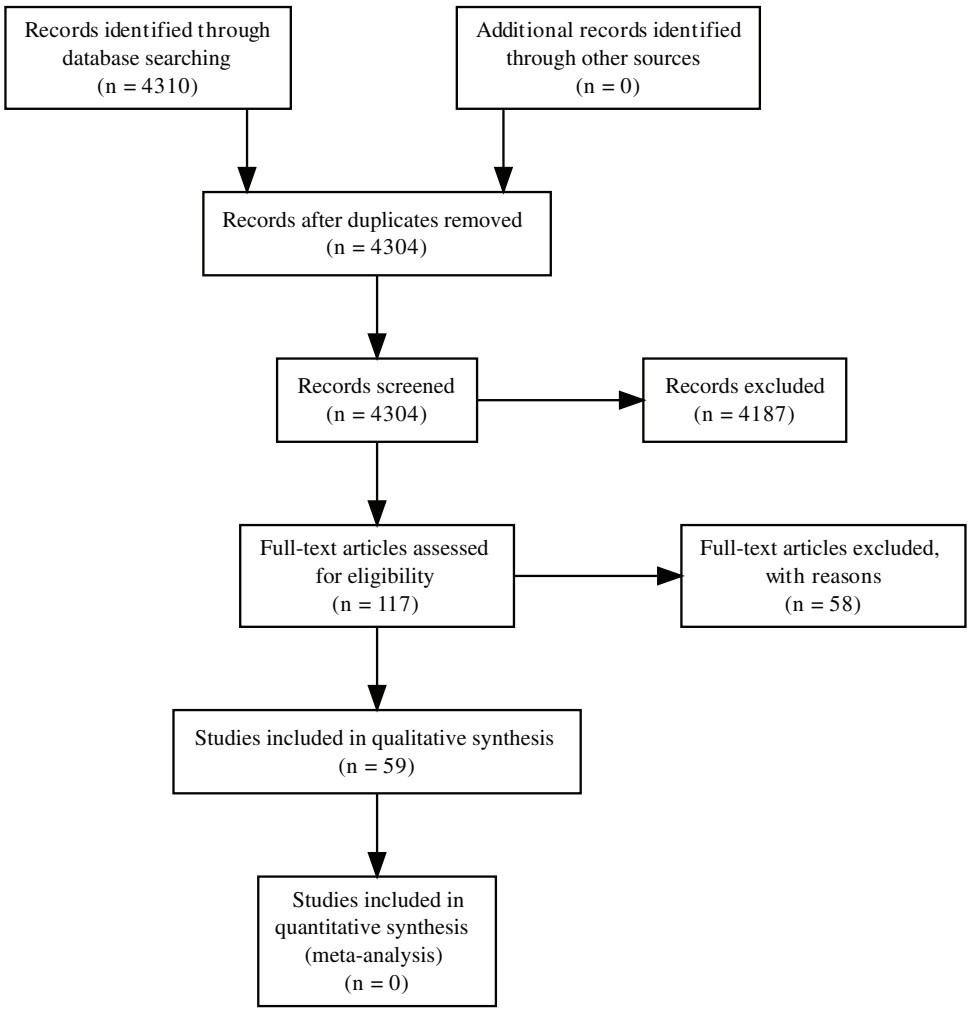

**Figure 1** PRISMA diagram for the progression of papers included in the analyses.

negative to positive (*García-cervigón et al., 2016*; *Pilosof et al., 2017*). However, the sum of the direct interactions between two species do not represent the net outcome of the relationship since they are non-additive; these direct interactions can in turn interact (*Proulx, Promislow & Phillips, 2005*). When each species interacts with at least one third party species, indirect interactions quickly occur (*Borrett, Whipple & Patten, 2010*). It is the sum of these direct and indirect interactions that represent the net outcome of the interaction between any two species (*Michalet et al., 2015*).

For most angiosperms, interactions with herbivores and pollinators impact fitness. Herbivory can be generally classified as having a direct negative effect on plants, while pollinators can be similarly classified as having a direct positive effect. Typically, herbivory and pollination are examined one at a time; however, these effect pathways frequently co-occur and therefore interact and so the net outcome is not necessarily as straightforward as these simple classifications (*Strauss, Conner & Rush, 1996*; *Vulliamy, Potts & Willmer, 2006*; *Tsuji et al., 2016*; *Chalcoff, Lescano & Devegili, 2019*;
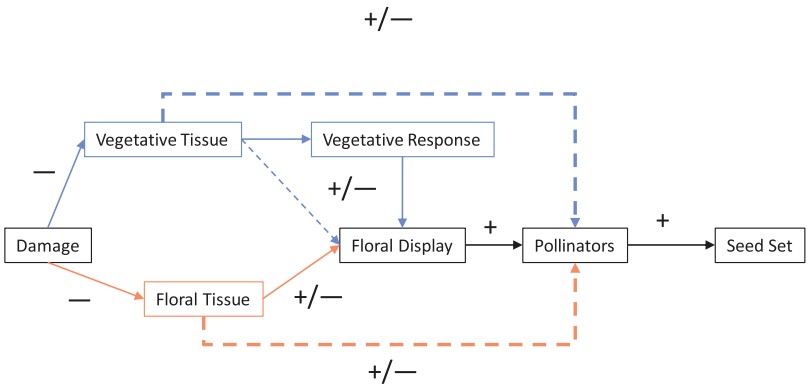

**Figure 2** **Mechanisms of damage by herbivores that can impact pollination and therefore seed set.**
Solid lines represent direct interactions and dotted lines indirect interactions. The two main pathways
are direct (direct damage to floral tissue influences pollinators; shown lighter in orange) and indirect
(damage to vegetative tissue indirectly effects floral traits; shown darker in blue). Lines and boxes in black
represent interactions and steps shared by both pathways. The dotted lines represent the net indirect
interaction of plant damage on pollinators (and pollination) that was the focus of this review.

*Rusman et al., 2019*; *Scopece, Frachon & Cozzolino, 2019*). In this systematic review (Fig. 1),
we have outlined a conceptual framework (Fig. 2) to illustrate the direct and indirect ways
in which the negative effects of herbivory can both directly and indirectly effect plant
fitness via animal pollinators and pollination. Here, we outline the general ways in which
different types of herbivory can impact plant fitness and how this interacts with pollination
as outlined in Fig. 2, followed by a synthesis of the contemporary literature on
herbivore–pollinator interactions.

Herbivory alone can impact plant reproduction both directly and indirectly, regardless
of pollination. Partial florivory (damage to floral tissue) or complete florivory (complete
removal of flowers; see Table 1 for a list of definitions) can reduce plant fitness by
directly reducing the capacity of a flower to produce seeds. However, herbivores can also
remove or damage non-floral (vegetative) structures such as leaves (folivory), stems (stem
damage), and roots (root herbivory). Damage to these structures can cause a plant to
either not produce flowers, fruits or seeds, or produce structures of poor quality
(e.g., non-viable seeds; *Brody, 1997*; *Mothershead & Marquis, 2000*; *Lucas-Barbosa, 2016*;
*Rusman et al., 2019*). Regardless of any indirect interactions via pollinators herbivory can
result in a negative net outcome on plant fitness.

The results of these types of damage influence floral display and therefore pollinator
visitation. Removal of flowers not only eliminates a potential source of resources for
pollinators, but also decreases the overall size of the floral display. Rather than removing
flowers, partial florivory can make flowers directly less attractive to pollinators by reducing
symmetry (*Botto-Mahan et al., 2011*). Partial florivory can also have indirect effects on
floral traits and pollinator attraction similar to consumption of vegetative (non-floral)
tissues (discussed below) such as reduction in flower size and nectar production (*Krupnick,
Weis & Campbell, 1999*; *Mothershead & Marquis, 2000*).
**Table 1 Definitions and study counts of all types of herbivory as well as floral, pollinator, and plant responses included in this review.** Study counts include artificial herbivory versions for each herbivory category (e.g., both floral herbivory done by animals and human removal of petals would be included under florivory).

| Term | Definition | Category | Number of studies |
|---|---|---|---|
| Florivory | Flower consumption, including removal of flowers and inflorescences (complete florivory) and partial removal of flowers and petals (partial or incomplete florivory) | Floral herbivory | 29 |
| Folivory | Leaf consumption | Vegetative herbivory | 16 |
| Stem damage | Damage to the stem, including puncture damage and meristem removal | Vegetative herbivory | 5 |
| Root herbivory | Damage to or consumption of roots | Vegetative herbivory | 3 |
| Open (herbivory) | Open to all herbivores that could consume any or all plant tissues | Both vegetative and floral herbivory with unknown proportions | 3 |
| Grazing | Indiscriminate consumption of plants by mammalian herbivores | Both vegetative and floral herbivory with unknown proportions | 12 |
| Flower morphology/ architecture and size | Refers to flower symmetry (both due to faulty growth and partial floral damage), inflorescence shape, and architecture, as well as aspects of floral morphology relating to size including diameter, surface area, and corolla length | Floral response | 16 |
| Flowering phenology | The timing of flowers, including when flowers are produced and when they open | Floral response | 5 |
| Flower abundance | The number of flowers in total. Also includes the presence/absence of flowers when flowers are considered individually | Floral response | 24 |
| Sex ratio | The relative proportion of male and female flowers | Floral response | 1 |
| Floral diversity | Number of species or other diversity metric of flowering species | Floral response | 2 |
| Pollen production | The amount of pollen produced by a flower or stigma | Floral response | 5 |
| Pollen deposition | The amount of pollen deposited by a pollinator | Pollinator effect | 6 |
| Pollinator visitation | The frequency with which a flower or plant is visited by pollinators | Pollinator effect | 35 |
| Pollinator abundance | The abundance of pollinators found in the local environment | Pollinator effect | 4 |
| Pollinator diversity | The number of pollinator species (or other diversity metric) that either visit a flower/plant or are found in the local environment | Pollinator effect | 3 |
| Fruit set | A number of measures that represent the amount of fruit produced including number of fruits, fruit size, and fruit mass | Plant response | 26 |
| Seed set | A number of measures that represent the amount of seed produced including number of seeds, seed size, and seed mass. In some cases, only viable seeds are considered | Plant response | 29 |

Vegetative herbivory such as folivory, root herbivory, and stem damage (Table 1) can indirectly influence floral display (similar to partial florivory). They can cause plants to produce fewer and smaller flowers (*Strauss, Conner & Rush, 1996*; *Hambäck, 2001*; *Hladun & Adler, 2009*) as well as change the morphology of flowers (including symmetry or architectural structure) (*Strauss, Conner & Rush, 1996*; Table 1; *Mothershead & Marquis, 2000*; *Suárez, Gonzáles & Gianoli, 2009*). Phenology, sex ratio and pollen production can further shift with vegetative herbivory (*Strauss, Conner & Rush, 1996*; *Mothershead & Marquis, 2000*; *Avila-Sakar, Simmers & Stephenson, 2003*; *Arceo-Gómez, Parra-Tabla & Navarro, 2009*). In these ways, both vegetative and floral damage can interact with pollination.

While herbivory is inherently negative and can have negative impacts on pollination, the net outcome of herbivory both on pollination and plant reproduction is not necessarily negative. The net outcome is mediated by plant responses (*Santangelo, Thompson & Johnson, 2019*) in terms of resistances, allocation strategies, and defenses (see *Hawkes & Sullivan, 2001*). Plants can overproduce in preparation for herbivory or compensate for herbivory received by producing more structures or switching to self-pollination, reducing the overall impact (*Garcia & Eubanks, 2019*). Plants can also deter herbivores through defenses including constitutive (e.g., thorns) or inducible (e.g., volatile release) defenses that can be both mechanical or chemical (*Chen, 2008*). Plants may also reduce herbivory by interacting with other non-herbivore species such as other plants (*Ruttan & Lortie, 2013*) and predators (*Heil, 2008*). By reducing herbivory or mitigating the damage caused by herbivores, plants are able to reduce their impact on pollination and plant reproduction. However, a cost to some defenses include deterring pollinators (*Lucas-Barbosa, Van Loon & Dicke, 2011*).

Numerous mechanistic pathways can integrate the direct and indirect impacts of herbivores on plant reproduction through plant tissue, allocation strategies, and timing that impact plant pollination (*Strauss, Conner & Rush, 1996*; *Mothershead & Marquis, 2000*; *Kelly et al., 2008*; *Botto-Mahan et al., 2011*). In turn, negative impacts to pollinators can amplify the negative effects of herbivores on plant fitness by reducing both potential seed set (e.g., number of flowers available to set seed) (*Strauss, Conner & Rush, 1996*; *Hambäck, 2001*; *Rusman et al., 2019*) and actual seed set (i.e., flowers are not all pollinated due to decreased pollinator visitation) (*Adler, Karban & Strauss, 2001*; *Benning & Moeller, 2019*). The nature of how not only each type of herbivory, but also the joint impact of multiple types of herbivory impact pollination and plant reproduction are the basis of Fig. 2.

In this systematic review, we synthesize the contemporary literature on herbivore–plant–pollinator interactions with a specific focus on studies that examined the joint impact of herbivores and pollinators on plant reproduction or the impact of herbivores on pollination using the mechanistic pathways proposed in our conceptual framework (Fig. 2). The frequency of mechanisms tested and the frequency that direct vs indirect floral damage pathways are contrasted is important to both ecology and evolution. This includes examining the diversity of types of damage—both the tissue targeted and the taxa causing the damage. Finally, we examine how each mechanism is tested.

## SURVEY METHODOLOGY

A search for papers that examine the impact of herbivores on pollinators or the pollination of plants in October 2019 using Web of Science and the search terms "herbivor* AND pollinat*", "floriv*", "foliv* AND pollinat*", "herbivor* AND flower*", and "foliv* AND flower" was conducted by S. Haas (no review protocol was registered). This resulted in 4,304 unique papers (Fig. 1). Papers had to meet the criteria that they directly tested the impact of herbivory on animal-mediated pollination. The indirect effect of herbivores on pollinators or the indirect effect of herbivores on plants via pollinators must have been reported to be included in this synthesis (e.g., through measuring pollen deposition or comparing open pollination to supplementary hand pollination). After review, 59 papers
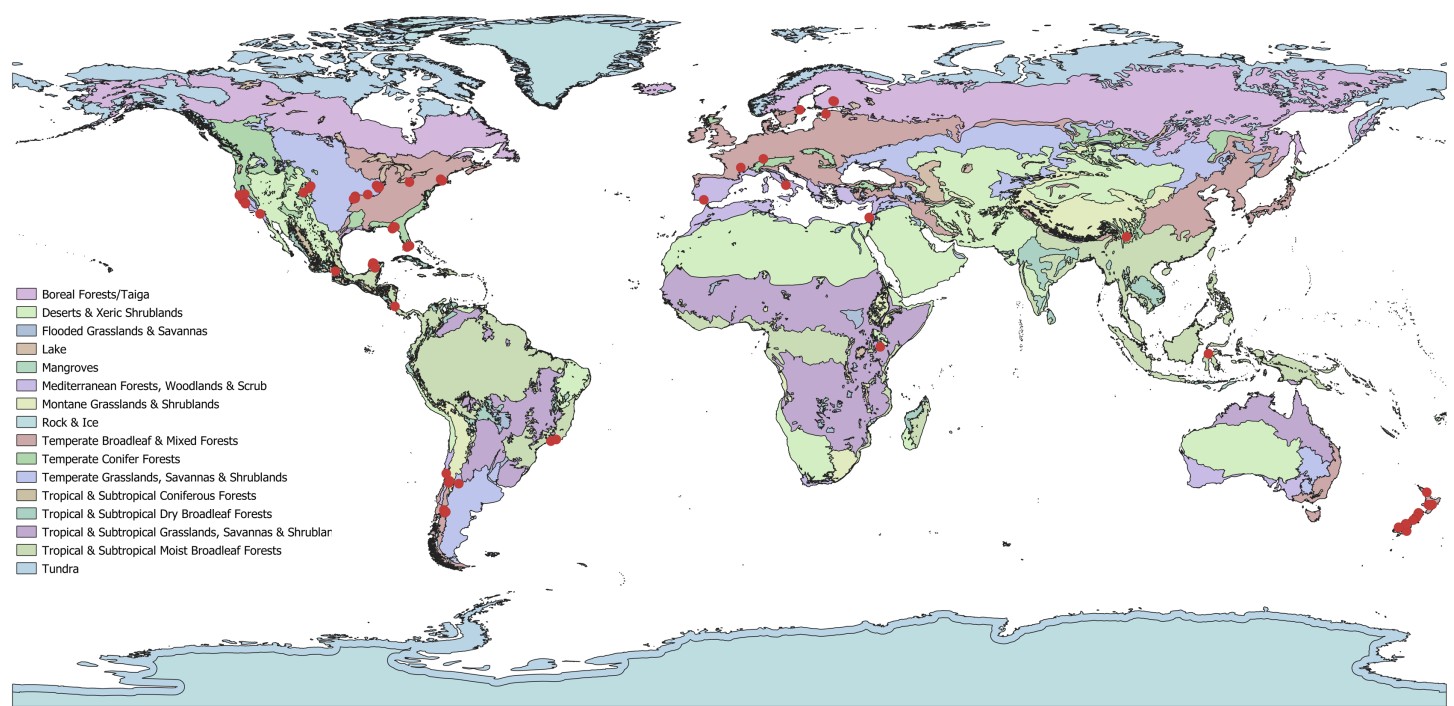

**Figure 3 Geographical distribution of studies (*N* = 56; 3 studies had no geographic information) across biomes that examined the impact of both herbivores and pollinators.** Studies spanned over 20 countries and 11 biomes. Biomes were generated from Terrestrial Ecoregions of the World, originating from the World Wildlife Fund (*Olson et al., 2001*). This figure incorporates data from the Terrestrial Ecoregions of the World database which is © World Wildlife Fund, Inc. (2006–20__) and has been used herein under license. WWF has not evaluated the data as altered and incorporated within the figure, and therefore gives no warranty regarding its accuracy, completeness, currency or suitability for any particular purpose.

were included in the final analysis (Fig. 1). Papers that were excluded were reviews and descriptions of the natural history of plants or animals (including diet). Studies were also excluded if they examined the impact of herbivores on plants but not pollination. Studies had to specifically test some effect pathway from herbivores to pollinators, studies that included both herbivores and pollinators but examined their effects on plants independently or examined the effect of some other factor on each group were not included. Studies on other types of consumption, such as nectar robbing, gall-forming insects, seed predation, and frugivory (consumption of fruits) were excluded. Plants also had to be animal-pollinated (at least in part). Data extracted included the physical location of all study sites and the taxa examined, as well as the analyses performed (including type of herbivory, response variable, and general direction of effect each variable had on each response) and the general structure of the experimental design. Site biomes were calculated using biomes from Terrestrial Ecoregions of the World, originating from the World Wildlife Fund (*Olson et al., 2001*) using the software QGIS (*QGIS Development Team, 2019*).

## RESULTS

In total, 59 papers met all criteria to be included in the final analysis. These papers ranged from 1995 to 2019 spanning 18 different countries and 11 (of 14) different biomes (Fig. 3). The majority were done in the United States and the temperate broadleaf & mixed
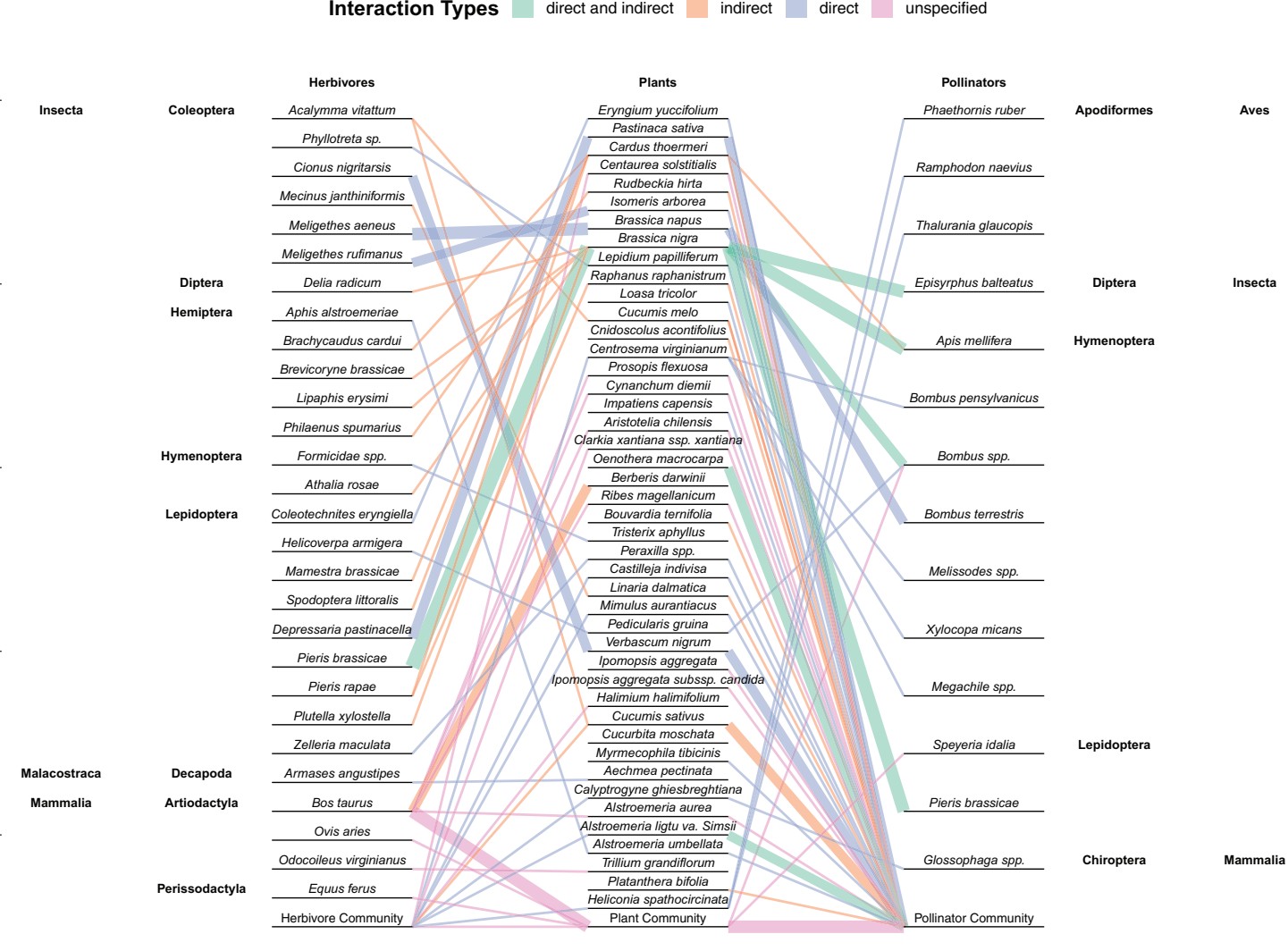

**Figure 4** Network showing the interactions between herbivores, plants, and pollinators found within the 59 studies included in this review.
Links are colored by whether herbivores had a direct, indirect, or unspecified effect (or both direct and indirect) on floral tissue within the study. Line thickness represents multiple interactions between those two species. Community refers to studies where the herbivores, plants, or pollinators consisted of whatever species were found within the natural community and not restricted. Plant species in which no herbivores or pollinators were used within the study (e.g., herbivory was artificially mimicked and pollination was measured passively through hand pollination) are not included.

forests biome. The only biomes not represented were tropical & subtropical coniferous forests, tropical & subtropical grasslands, savannas & shrublands, and tundra. Of the 56 papers in which site information could be taken (that were not greenhouse experiments) all but three studies were located within a single biome. A total of 47 plant taxa, 27 herbivore taxa, and 18 pollinator taxa were studied in these papers (Table S1; Fig. 4). Almost all herbivores (81%) and pollinators (78%) were insects. A total of 90% of studies ($N = 55$) examined a single plant species while only 43% of studies ($N = 26$) examined a single herbivore and 10% a single pollinator ($N = 6$; Tables S2 and S3). It was most common to examine the entire community of pollinators (72% of studies; $N = 44$).

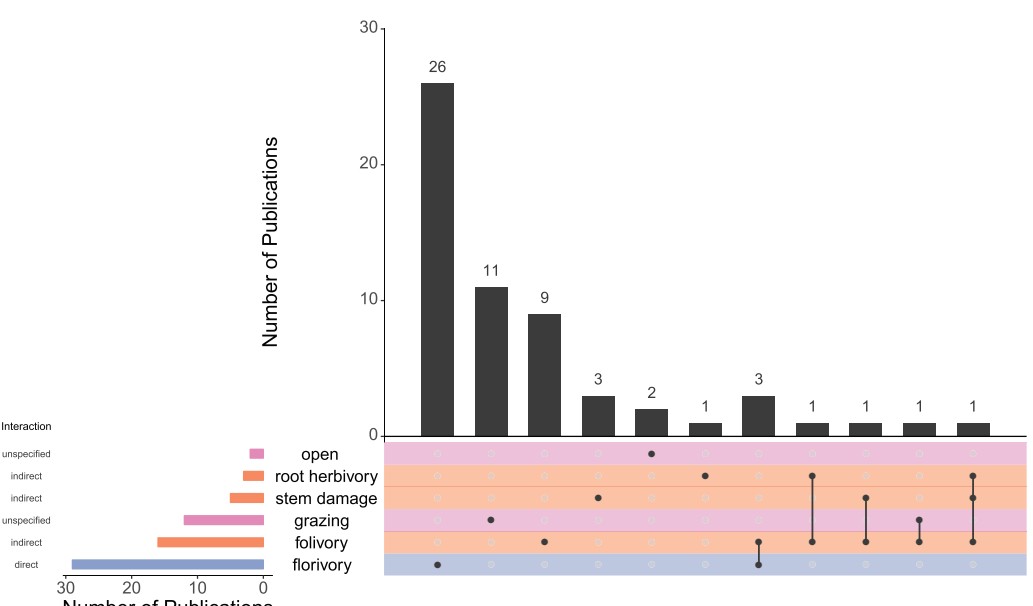

**Figure 5 UpSet plot showing the number of publications (N = 59) that examined each type of herbivory both individually and simultaneously.** The black vertical bars represent the number of publications that looked at exactly one, two or three types of herbivory. The dots directly below black bars correspond to which type(s) of herbivory are represented within that category. The colored horizontal bars to the left of the list of herbivory types show the number of publications that included each herbivory type (regardless of whether another type of herbivory was also examined). Horizontal bars and rows are colored according to the interaction type of each form of herbivory (i.e., direct, indirect, or unspecified). Unspecified interaction pathways are those in which one or both of direct and indirect pathways are possible, but not specified.               

Pollination was most frequently measured through direct pollinator visitation (57% of studies); however, 25% used supplemental pollination (Table 1). Herbivory was also most frequently (67%) observed directly by animals although 33% of studies ($N = 20$) applied some form of artificial herbivory, and 22% artificially reduced herbivory by excluding herbivores or applying pesticides.

The most common type of herbivory examined was florivory (49%; Fig. 5; Table 1), with every one of these studies examining partial florivory and only one also examining complete florivory. This was followed by folivory (27%) and grazing (20%). However, stem damage (8%) and root herbivory (5%) were also utilized. For 5% of studies, herbivory was non-specific (i.e., "open" to all herbivores). Almost all studies (88%) looked at only one form of herbivory. Of the other 12% of studies, two or three types of herbivory were examined. *Hladun & Adler (2009)* examined the interaction between two types of vegetative herbivory—root herbivory and folivory. *Buchanan (2015)* also compared two types of vegetative herbivory: leaf damage and meristem damage, while *Rusman et al. (2019)* looked at all three types of vegetative herbivory (folivory, stem damage, and root herbivory). Similarly, *Sasal, Farji-Brener & Raffaele (2017)* looked at both general grazing (in the form of ungulates) and specifically folivory (in the form of insect herbivory). *Lucas-Barbosa et al. (2013)*, *Lucas-Barbosa et al. (2016)* and

**Table 2 Distribution of studies based on herbivore-pollinator mechanism examined.** Unspecified refers to damage that may include either of vegetative or floral tissue or both.

| Indirect | n | Direct | n | Unspecified | n |
|---|---|---|---|---|---|
| Vegetative damage-floral response-pollinator | 5 | Floral damage-floral response-pollinator | 1 | Unspecified damage-floral response-pollinator | 6 |
| Vegetative damage-floral response-pollinator-seed set | 9 | Floral damage-floral response-pollinator-seed set | 6 | Unspecified damage-floral response-pollinator-seed set | 2 |
| Vegetative damage-pollinator | 3 | Floral damage-pollinator | 7 | Unspecified damage-pollinator | 0 |
| Vegetative damage-pollinator-seed set | 4 | Floral damage-pollinator-seed set | 13 | Unspecified damage-pollinator-seed set | 6 |

*Mothershead & Marquis (2000)* compared the effects of florivory and folivory representing the only papers that compared vegetative and floral herbivory.

Florivory was found to have some negative impact on floral expression, pollination, or seed production in 86% of papers (Table S4). Some positive effect of florivory was found in 24% of papers and a neutral effect in 48% of papers. For folivory, 81% of papers each found some negative effect or neutral effect, while only 31% found any positive effect. Grazing had 67% of papers each find some negative or neutral effect, and 33% found some positive effect. Similar proportions were found in each of the other types of herbivory.

Only 19% of papers took a full mechanistic approach to the effects of herbivores on pollinators (Table 2). These studies examined each point of the mechanism in Fig. 2: the effects of herbivory on floral display, pollinators, and plant reproduction. Most frequently (24%), papers examined the effect on pollinators and plant reproduction while skipping the mechanistic step of the impact on the flower. Otherwise, there was an approximately equal split between only examining effects on pollinators (15%), only examining the effect of supplemental pollination on reproduction (12%), examining the floral attributes and reproduction without the pollinator (15%), or the floral attributes and pollinator without reproduction (15%).

## DISCUSSION

Herbivory and reproduction in plants are intimately linked through interactions with animals. In this systematic review, we examined the relative frequencies and the extent that these important processes are studied in concert. While the effect of herbivores on plants have long been studied, and it has been well shown that herbivores can directly and indirectly impact plant reproduction, growth, and population dynamics (*Hawkes & Sullivan, 2001*; *Ohgushi, 2005*; *Boivin, Doublet & Candau, 2019*; *Garcia & Eubanks, 2019*), the small number of studies that were located within this review indicate how infrequently the effects of herbivores on pollination are studied. These studies were also heavily biased towards damage done by insects, temperate biomes, and the effects of a single herbivore species on a single plant species. Furthermore, only a handful of studies compared direct and indirect effects of herbivory on floral traits and pollination. Given that most animal-pollinated plants likely experience damage to multiple tissues as well as pollination (*Lucas-Barbosa, 2016*), this is an unfortunate gap in the literature. In addition, those few studies that do examine both herbivory and pollination frequently examine

only the net outcome and not the underlying mechanism (i.e., how herbivory impacts floral traits and how floral traits impact pollination). That is, most studies do not fully examine the interaction network outlined in our framework (Fig. 2). Examining the impacts of multiple herbivores as well as multiple types of herbivory is important in determining how plants and pollinators are impacted by real communities of herbivores.

## Effects of florivory on pollination

Direct damage to floral tissue is an important factor in determining plant reproductive output. Complete florivory has been shown to have direct impacts on floral abundance. The net outcome of this form of direct damage is dependent on the strategy of the plant (*Juenger & Bergelson, 1997*; *Wise, Cummins & De Young, 2008*; *Garcia & Eubanks, 2019*). In some cases, plants will over-produce flowers as a defense against florivory creating more flowers than the plant is able to bring to seed (*Huth & Pellmyr, 1997*). Therefore, florivory frequently results in no net loss in reproductive output for the plant. In other cases, plants are able to compensate or overcompensate for herbivory, replacing the flowers lost (sometimes producing more flowers than initially), delaying phenology (*Wise, Cummins & De Young, 2008*; *Garcia & Eubanks, 2019*), or changing mating system (i.e., shifting to self-pollination) (*Penet, Collin & Ashman, 2009*). If the plant is able to completely compensate (reproduce flowers of equal or greater quality and quantity) then the net effect of complete florivory is neutral or even positive. How plants interact with florivores can influence communities and population resilience through these differences in reproductive output. While the impact of removal of flowers on overall plant reproduction has been studied numerous times (*McCall & Irwin, 2006*), the impact of this removal specifically on pollination or pollinator visitation is rarely studied. Out of the 29 papers that examined the impact of florivory on pollination found in this review, only one examined complete florivory (*Sutter & Albrecht, 2016*). Changes in plant population and community dynamics due to changes in reproduction have the potential to impact pollinators; for instance, patches with more flowers tend to attract more pollinators (*Lazaro & Totland, 2010*). Pollinators in turn interact with plants to determine reproductive output. Therefore, the indirect interactions between herbivores and pollinators fosters even further co-evolutionary processes such that plants not only sufficiently compensate for lost reproductive structures due to herbivores, but also to produce flowers of quality and quantity sufficient to attract pollinators. This interaction requires further research into the implications of complete florivory on plant compensation, pollination, and reproduction.

Incomplete florivory can also impact both pollination and plant reproduction. Incomplete florivory can result in flowers that are less attractive to pollinators despite offering the same reward (*Mothershead & Marquis, 2000*). Pollinators can use visual cues such as floral symmetry to choose between flowers (*Rodríguez et al., 2004*). A loss of symmetry can result in decreased visitation (*McCall, 2010*). When other cues are more important, there may be no effect of incomplete florivory (*Malo, Leirana-Alcocer & Parra-Tabla, 2001*) and plants can mitigate or eliminate the negative effects of herbivory

on floral display by reproducing via self-pollination (with or without pollinators) in some cases including several species in this systematic review (*Cardel & Koptur, 2010*). While this review focused on animal-pollinated plants, many plants are not wholly reliant on animals for reproduction (*Culley, Weller & Sakai, 2002*). However, since animal pollination frequently increases plant fitness (*Klein et al., 2007*; *Cardel & Koptur, 2010*; *Jorge, Loureiro & Castro, 2015*) this strategy may only limit the effects of incomplete florivory as opposed to eliminating them. In addition, the actual presence of florivores in flowers can deter pollinators. For example, *Canela & Sazima (2003)* found that florivorous crabs not only decreased attraction of flowers to pollinators through damage but that pollinators were less likely to visit flowers while the crabs were present. All of the 29 studies we found that examined the impact of florivory on pollination examined partial florivory. In most studies, partial florivory was found to decrease pollinator visitation or pollen deposition as well as plant reproduction (fruit set or seed set). By decreasing pollinator visitation, incomplete florivory can indirectly decrease plant reproduction (via pollen limitation). As with complete florivory decreases in reproduction can impact population dynamics, while indirect effects on pollinators can drive the coevolutionary arms race between herbivores and plants that might not otherwise occur under the limited damage of incomplete florivory (that is, florivory that keeps ovules and stigmas intact).

## Effects of vegetative herbivory on pollination

While direct damage to floral tissue is the most common way to examine the effects of herbivores on pollinators (Fig. 4), damage to vegetative tissue also had indirect effects on floral attributes. The main mechanism that folivory, root damage and stem damage impact pollinators is through decreasing both resources and the ability for plants to produce resources (*Mothershead & Marquis, 2000*). By decreasing the amount of photosynthetic and absorptive area available to a plant or siphoning off xylem or phloem, fewer or smaller flowers may be produced (*Mothershead & Marquis, 2000*; *Hambäck, 2001*; *Hladun & Adler, 2009*). These flowers may be less attractive to pollinators (*Mothershead & Marquis, 2000*) or theoretically be less fertile, producing fewer seeds. While plants are also able to compensate for vegetative damage, resources are often allocated to regrowth instead of reproduction and so vegetative damage can still decrease fitness (*Pratt et al., 2005*; *Garcia & Eubanks, 2019*). Root herbivory can also change how the plant interacts with aboveground herbivores and mutualists (*Barber et al., 2015*). For instance, root herbivory can decrease aboveground herbivory and increase the nectar in extrafloral nectaries (*Hladun & Adler, 2009*; *Soler et al., 2012*).

While folivory, root herbivory, and stem damage can decrease reproductive output (*Mutikainen & Delph, 1996*; *Lehtilä & Strauss, 1999*; *Pratt et al., 2005*; *Lopez-Toledo et al., 2018*), it is less clear whether they impact pollinators or pollination. Folivory, root damage, and stem damage were found to negatively impact several floral traits, as well as pollinator visitation and reproduction (*Mutikainen & Delph, 1996*; *Strauss, Conner & Rush, 1996*; *Mothershead & Marquis, 2000*; *Hambäck, 2001*; *Arceo-Gómez, Parra-Tabla & Navarro, 2009*; *Hladun & Adler, 2009*; *Barber & Gorden, 2013*; *Sasal, Farji-Brener & Raffaele, 2017*). Folivory was found to have negative effects on most floral traits including

floral morphology, abundance, and phenology. Stem damage was found to have a negative effect on floral morphology, size, and abundance while root herbivory affected floral abundance and pollen production. However, the number of studies that found each of these effects is low and each type of vegetative damage was also frequently observed to have no effect on each of these respective traits and occasionally a positive effect. It is also possible that plants are better able to compensate for or resist vegetative damage such that there will be no change in floral display or reproduction. For instance (as with incomplete florivory) plants may switch to self-pollination if floral display is compromised or pollination is limited.

While it is clear that vegetative damage can impact pollination and plant reproduction, vegetative damage was also frequently observed to have no effect. This lack of effect may be only representative of small sample size and more studies would find the proportions more similar to what is found with florivory. However, finding fewer studies may be because few studies examined different types of herbivory (Fig. 5) or different taxa simultaneously (Fig. 4; Table S3). The same herbivore can feed on multiple tissues (at the same time or switching ontogenetically; e.g., *Lucas-Barbosa et al., 2016*) or multiple herbivores can feed on different tissues (or even the same tissue) simultaneously (*Barber, Adler & Bernardo, 2011*). Therefore, it is difficult to determine whether different types of herbivory may act synergistically or if they interfere with each other (as seen between root herbivory and aboveground herbivory in *Barber et al. (2015)*). The larger proportion of neutral effects of vegetative herbivory on pollination may only be an indication of not considering damage to all types of vegetative tissues.

This lack of directly comparing individual species of herbivores is one weakness of some of the papers included in this study. While comparing the effects of a broader taxonomic scope or community of herbivores or pollinators is good for comparing the net outcomes of interactions, the exact effects and net outcomes of individual species is lost. More research that examine specific species, especially multiple specific species could help illuminate these differences. This is particularly prevalent with how few studies examined individual pollinator species compared to those that studied the entire community (Fig. 4). In contrast, the indirect pathway from vegetative damage to changes in pollination may simply be more heavily regulated by plant physiological responses with plants preferentially allocating resources to reproduction over regrowth (Fig. 2). Considering multiple species in this case may not change this result. Regardless, the small sample size makes any conclusions about the relative proportion of studies to find significant or neutral effects of vegetative damage dubious.

## Integrating the effects of floral and vegetative herbivory

Vegetative herbivory can impact plant populations and communities through plant reproduction, but the role of the indirect effect of vegetative herbivory on pollinators and the role of pollinators in driving co-evolution between plants and non-floral herbivores is less distinct than when examining florivores. In order to determine the relative effect of direct and indirect damage to floral tissue on pollination, these two mechanisms need to be compared more frequently. In this systematic review, only three

studies examined the direct and indirect effects of herbivory on floral display and pollination (Fig. 5). Specifically, these three studies compared florivory to folivory. *Lucas-Barbosa et al. (2013)* examined the behavior of pollinators of *Brassica nigra* in response to the specialist caterpillar *Pieris brassicae*. *Pieris brassicae* feeds on the leaves of *B. nigra* at a younger stage, and progresses later to consuming flowers. Therefore, while examining damage to two types of tissues, the damage was done by the same individuals. They found there was no effect of *P. brassicae* on pollinators during the folivory stage, while there was an effect at the florivory stage. In a study with the same system by many of the same authors (*Lucas-Barbosa et al., 2016*) where the effect of damage to vegetative and floral tissues on floral volatiles detected by pollinators was studied, neither folivory nor florivory influenced pollinators.

Finally, *Mothershead & Marquis (2000)* examined the effect of artificial damage to both leaves and buds to the floral traits and seed set of *Oenothera macrocarpa* in the presence and absence of supplemental hand pollination. Both folivory and florivory affected floral traits (both morphology and size), that in turn impacted pollination and seed set. Folivory was not found to directly reduce seed set through reduced floral resources, but rather only indirectly through floral morphology. However, floral damage decreased fruit set (68% reduction) more than foliar damage (18% reduction). While two of these studies point towards florivory having a greater impact on pollination than folivory, two of three studies is not sufficient sample size to determine the relative importance of direct (florivory) over indirect (vegetative herbivory) damage. Only multiple studies that directly compare florivory and other types of herbivory within the same system will be sufficient to determine their relative importance.

Similarly, the larger proportion of papers that examined florivory over other forms of herbivory (Fig. 5) or the greater proportion of studies with a negative impact on pollination or reproduction due to florivory is not sufficient to make the claim that florivory has a greater impact on pollination than damage to vegetative tissue. Florivory is the more obvious choice when studying the effects of herbivory on pollination and so a bias in papers towards florivory is expected. Similarly, the sample size of studies that examine any other form of herbivory is particularly low, and so proportions are not necessarily representative. While it is intuitive and may be true that direct damage to floral tissue has a greater impact than indirect damage on pollination, there is not sufficient evidence to make this claim.

Some herbivores act as both herbivores and pollinators at different ontogenetic stages (*Lucas-Barbosa et al., 2016*). This type of switch is commonly seen in pollinating insects with a herbivorous larval form (*Nakazawa, 2015*). However, only a single species, the *B. nigra* specialist *P. brassicae*, was examined as both a herbivore and a pollinator. Furthermore, *P. brassicae* was only examined as both herbivore and pollinator of *B. nigra* in a single study (*Lucas-Barbosa et al., 2016*), although it was also used as a herbivore in *Lucas-Barbosa et al. (2013)* and as a pollinator in *Rusman et al. (2019)*. The intricate relationship and co-evolution between species that change between negative and positive interactions is not one that is unstudied (see *Nakazawa, 2015*). Strategies that reduce the impact of herbivores at an early stage that might negatively impact the later production

of floral resources would be beneficial to both plant and herbivore in this case, even more so than with species that do not share this relationship. However, clearly the net outcome of early stage herbivory on plant reproduction and late stage pollination is lacking within the literature.

### Effects of damage to unspecified tissue on pollination

The joint impact of damage to multiple tissues can be extended to the effects of herbivores that do not have a plant tissue preference. Most studies examined damage to specific tissues; however, a number of studies examined damage to unspecified tissues, representing both direct and indirect mechanisms that are not differentiated. Grazing encompasses possible damage to flowers, leaves and stems. Grazing is of particular import because of its potential severity and anthropogenic causes. The agricultural industry plays a large role in the impact humans have in creating disturbed ecosystems (*Kitzes et al., 2008*). Most studies that examined direct or indirect damage to floral tissue used insects as focal herbivores; those that looked at unspecified damage exclusively used mammals (Fig. 4). While studies frequently look at grazing by large mammals such as deer and cattle at a community level—examining the plant, or even floral diversity of a system (*Olff & Ritchie, 1998*; *Kohyani et al., 2008*; *Herrero-Jáuregui & Oesterheld, 2018*), it is rare for these studies to further examine the pollination consequences of grazing. Studies that examined grazing reported some negative effect of grazing on plant reproduction or pollinator visitation. Grazing also impacted floral morphology, abundance, phenology, and pollen production, but the mechanisms were not clearly reported in primary studies. By studying the synergistic effects of multiple effect pathways, we can better understand how grazing can impact vegetation.

### Other interactions

While the indirect effects of herbivory on pollination are the focus of this review; there are other ways in which pollinators and herbivores interact. For instance, there are direct interactions between herbivores and pollinators where the presence of herbivores actively deters pollinators from approaching flowers (*Canela & Sazima, 2003*). Additionally, pollination may impact herbivory by facilitating the successful reproduction of herbivore plant hosts. However, these types of interactions are neither included in this review nor the conceptual framework.

## CONCLUSIONS

Both direct and indirect damage to floral tissue can impact pollination and plant reproduction. However, direct and indirect damage to floral tissue is rarely examined in concert, nor are damage by different herbivores. The relative importance of the direct and indirect mechanisms and synergistic effects of multiple herbivores have important implications for ecological resilience and stability in evolutionary processes. However, this relative importance is almost never examined with the focus lying on each type and each herbivore individually. The indirect effect of herbivores on pollinators can mediate co-evolutionary processes between plants and herbivores and plants and pollinators.

The collection of herbivores that interact with plants can include species that feed on all types of tissue either simultaneously or temporally separated that the plant then integrates into growth, allocation, defense, or phenology. This in turn can impact pollinators and pollination, making these two plant–animal interactions intimately linked.

### Funding
This work was supported by York University Faculty of Graduate Studies and the Natural Sciences and Engineering Research Council of Canada. The funders had no role in study design, data collection and analysis, decision to publish, or preparation of the manuscript.

### Grant Disclosures
The following grant information was disclosed by the authors:
York University Faculty of Graduate Studies and the Natural Sciences and Engineering Research Council of Canada.

### Competing Interests
Christopher J Lortie is an Academic Editor for PeerJ.

### Author Contributions
- Stephanie M. Haas conceived and designed the experiments, performed the experiments, analyzed the data, prepared figures and/or tables, authored or reviewed drafts of the paper, and approved the final draft.
- Christopher J. Lortie conceived and designed the experiments, authored or reviewed drafts of the paper, and approved the final draft.

### Data Availability
Data is also available on FigShare (*Haas & Lortie, 2020*).
The raw data are available in the Supplemental Files.

### Supplemental Information
Supplemental information for this article can be found online at http://dx.doi.org/10.7717/peerj.9049#supplemental-information.

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
