# Peer review of "A systematic review of the direct and indirect effects of herbivory on plant reproduction mediated by pollination"

_PeerJ, doi:10.7717/peerj.9049_

## Round 0.1 · original submission · Major Revisions

· Academic Editor

Major Revisions

The topic addressed in the manuscript has been evaluated by both reviewers. However, both reviewers agree the manuscript cannot be accepted in its current form. The article should be revised deeply, both in structure and organization. Authors should reorganize the work and consider the very valuable reviewer's comments, which I fully share. I encourage you to prepare a new version substantially improved in the view of all the comment suggestion find below.

Reviewer 1 ·

Basic reporting

This work reviews the literature regarding herbivory and pollination in order to evaluate their effects and influences over plant reproductive success. Despite that to know the gaps and research needs in the field is very necessary, this work do not sustain an original proposal and is lacking of the background needed to suffice as an original publication. Below I comment some points that describe this problem.

1-The tittle, “A systematic review of herbivore-pollinator interactions: theories, methods, and challenges”, do not reflect the content of this ms. This is not clearly about the “interactions of pollinators vs. herbivores” but rather is more focused the influences of herbivory over plant reproduction mediated by animal pollination. The whole work seems to be oriented to understand plant fitness consequences therefore, this must be clearly stated in the tittle.

2-Conclusion paragraph in the Abstract is vague and should provide the main message of this work in a more straightforward way. For instance, line 58-59 in the Introduction and lines 278-279 in Conclusions provides hits on the goals and main findings of this manuscript.

3-Figure 1 is incomplete and inaccurately represents the direct and indirect effects on plant reproductive success due to herbivory vs. pollination. Current figure is only considering the effects from animal foraging on vegetative and floral structures over a “flower” category box. This is not useful to understand the process as the effects of different stresses may influence plant physiology and development, triggering changes on reproductive budget over time that could be reflected in less gametes and floral resources (such as nectar, fragrances, oils etc.) as well as the overall quality of floral display. This figure does not allow to easily discussing these different effects.

4-Plant response towards pollination can also affect herbivory. Several organisms change the sign of its ecological interaction through ontogeny. For instance, many insects are first herbivores (folivores, florivores, gall-inducers, among others) and later, as adults, contribute to pollination. By not considering this connection you loose the chances to discuss on the lack of experimental studies evaluating this kind of ontogenetic switching between herbivorous and pollinator role of several animals and its plant fitness consequences. This is especially important in perennial species.

5-Moreover, a good portion of plant species present mixed mating systems where their reproductive strategy can be heavily influenced by: 1. Effective pollinator visits and 2. Herbivory. Plants may switch towards self-reproduction when they are stressed by the lack of visits by pollinators (that could be due to a loss of attractiveness by florivory) or due to a heavy pressure of herbivores. Your model cannot be applied to a considerable amount of plant species. In Table 3, where authors draw connections between plants and animals associated, there are plant species that are capable of producing seeds WITHOUT pollinators!, such as for example Tristerix aphyllus (Loranthaceae) is capable of asexual reproduction by selfing and even apomixis. This is the most serious drawbacks of this work as the ms. is only considering animal pollination as the source of plant fitness.

6-Introduction: This section is not very well written; authors should drive readers towards their goals. Mechanisms involved in plant response towards the interaction of animals are very well documented and should be presented using its nomenclature, in a more concise and clear way. For instance, instead of “pre-formed” defenses use “constitutive”. Furthermore, several stresses plants suffer due to herbivory depends on the kind of foraging behavior, taxonomic identity of the attacker and are expressed in varied temporal frames. All these must be explained and considered if this review intends to be systemic. Each of the definition related to this characterization must be provided in this section, or in Table 1 associating the intro with it, besides of categories of foraging attacks authors must define for instance: kind of plant responses (either resistances, defenses, etc.) timing of these responses, among others. For example “complete vs. incomplete florivory” is only mentioned later in discussion.

7-Regarding Methods, instead of using country frontiers for showing where the articles used belong. I recommend authors to use the information regarding the kind of biome where these researches were done. In this way they will be able to inform if there is a bias on the kind of habitat where these research have been done. Moreover, as mentioned above, the temporal frame considered between herbivory versus pollination effects must be considered in the analysis. Preponderance of herbivory damage as well as the abundance of pollinators can depend strongly on biome, therefore, this is an additional issue that is not addressed nor discussed in this draft.

8-JA manipulative experiments should not be classified as “herbivory” as these only considering some aspects of the reactions of plants towards a stress similar to herbivory damage.

9- Concerning the evaluation of papers to be used in this review it seems that there were no filter regarding an adequate taxonomic identification of the animals interacting with the plant species. The quality of animal identification is heterogeneous in the articles selected for this revision. This is not trivial as even organisms belonging to a same genus may display negative or positive interactions with plants. If a better selection of works based on taxonomic accuracy was not possible, due to the reduced number of works, at least this should be commented as an additional issue to be solve by future works.
Similar problems are found in the plant species lists, it would be desirable that plants species name be accompanied by their family classification. This provide additional information to discuss on potential research biases.

10-Discussion. Do not reflect the whole development of this work. For instance only Fig. 1 is referred in this section. Why Fig. 3 is not even mentioned here? There are no original conclusions. Most of what is discussed seems to be extracted from previous revisions such as Garcia & Eubanks, 2019 or Mothershead & Marquis, 2000. It is possible that the lack of meta-analysis, or any statistical tool, in this draft do not allow authors to conclude based on the data complied from this revision. In general this section do not provide new ideas to what has been already published in previous reviews on this topic.

I consider this work do not curated the information used for the revision adequately, and is far from a systemic review on the reproductive effects of herbivory and pollination over plant fitness. This ms. do take advantage of the massive body of evidences from plant reproductive ecology or chemical ecology approaches that otherwise may have provided a solid background to the bibliographic inquiry here presented. Unfortunately, this does not merit publication.

Experimental design

All comments in 1

Validity of the findings

All comments in 1

Reviewer 2 ·

Basic reporting

The organization and flow of the introduction and discussion were somewhat scattered and not always connected to the main point of the methods. Some of this has to do with larger potential issues (i.e. what the purpose of this paper is meant to be; see “General comments” below). But even setting that aside, the organization needs some work. Paragraphs are frequently long, and jump from topic to topic. For example, the first paragraph of the paper covers direct and indirect effects of herbivory, the importance of pollination, potential interactions between herbivores and pollinators, and effects on plant fitness; that’s enough information/topics for three or four paragraphs.

There’s also a few language choice problems, including inconsistent use of “direct” and “indirect” (sometimes stated clearly, other times assumed or stated unclearly) and the word “pathways” which seems to have at least three meanings in the introduction (pathways of effect between herbivores and pollinators, different kinds of herbivory, or different classes of interactions such as mutualism vs. antagonism). Also, the introduction uses the hypothesis in Figure 1 as part of the justification for the paper, but doesn’t ever directly talk about that figure or its implied hypothesis/es.

Raw data have been supplied and are mostly labeled well, though some of the values, abbreviations, and column headings are unintuitive. For example, using “number of pollinators” = “100” to denote an entire community was studied. The accessory documents define all of these well enough, but there may be changes that would make the data easier to understand without having to reference additional documents.

I like Figure 1, and I think it could serve as a hypothesis for a path analysis in future research. However, on two different computer screens the green color used for some arrows was very difficult to tell apart from gray or black; when printed in grayscale, it was even worse. Figure 4 is very information-dense, which makes it somewhat difficult to parse – but the figure legend explains things fairly well. Figure 3 was the figure I struggled the most with. I understand what it’s trying to show (and how), but the lines don’t always connect with the species names and it’s frequently hard to tell what the lines are connecting or pointing to. It’s also not obvious what order the interactions are listed in; it’s not alphabetical by species names.

The tables in both the paper and in the supplemental documents were hard to tell where horizontal dividers belonged. Additionally, text in each column isn’t necessarily justified vertically in the same way across rows. For example, in Table 1, the term “Root Herbivory” is justified higher in the cell than its definition in the next column, which makes it hard to read as a cohesive row. There needs to be either space between each row or horizontal lines added.

Experimental design

I had trouble deciding what the main purpose or question of this paper was meant to be; see my comments below in “General comments.” There was very little in the way of experimental design and even less about how the analysis was completed. The methods section was only one short paragraph long, and focus on how papers were found and sorted, with no information about what data were taken from those papers or how those data were further analyzed. Overall, methods and results focus mostly on qualitative information, with less interest in analytical or technical information.

Validity of the findings

It’s not always entirely clear how the authors decided which paper goes in which category, especially when it comes to things like simulated herbivory/florivory or studies with multiple species or studies that didn’t measure reproduction etc. I tried to parse through some of the data based on citations I’m familiar with, and frequently I would have put different data points in their table. This makes it difficult to know if the summary of what kinds of studies have or haven’t been published is accurate. I can’t tell if this is a matter of providing consistent, logical definitions for each data choice, or if the data itself are too subjective to be interpreted analytically.

The main conclusion of the paper seems to be that herbivores effect pollinators (directly and/or indirectly) which in turn can affect plant pollination success. As a conclusion, this feels weak; this is something that is already well known by ecologists in this field, and probably assumed by ecologists in related fields. Additionally, that conclusion doesn’t tie back to the results in this paper (looking at where more studies are needed, for example). This conclusion is valid, but it doesn’t serve either the purpose of the paper or the scientific audience of a paper like this.

Additional comments

I’ve read this paper several times through in preparing to write this review. This general topic is one of great interest to me. There have been previous reviews that touch on how herbivory and/or florivory specifically affect pollination and (potentially) plant reproduction, but they are all several years old now. I think there is room to have a review that talks about what we know and what we have left to learn on the topic. However, I’m not convinced this paper fills that role.

The methods and results of this paper focus on the number of papers published in this field, pointing out what kind of studies are most and least common. But the methods and results don’t attempt to summarize the kinds of results those papers found. Meanwhile, the introduction and discussion provide a cursory summary of the results from those papers (as you would find in the introduction section of an original research paper, for example) but doesn’t really put them in the context of how many articles of each type have been published. This makes the paper feel like two separate things, neither of which is completely satisfying.

I think the authors may need to think about what they’re trying to accomplish with this piece. If their goal is mainly to point out the gaps in terms of where papers are or are not being published, that can be done in a much more concise manner (a “Note” article, for example) that could summarize the information presented in the methods and results section of this paper as a sort of call to arms for researchers. This could be of interest to anyone familiar with the field, and might provide a jumping-off point for new research project ideas. However, an article of that type would need only limited review of results from those previous papers (since the “results” would be how many and what kind of articles are being published); in that case, the information currently in the introduction and discussion of this paper would be greatly reduced to a couple of paragraphs for context.

If the main goal with this paper is to review the results from previous herbivore-plant-pollinator studies, then there should be less focus on how many papers have been published and more focus on what the results of those papers were. This may or may not include a meta-analysis, if you want a more analytical approach to the data. Even a traditional review, however, could provide new insights into what the common trends are in this field, with room to point out remaining gaps in our knowledge. In a review paper of this type, the results and methods section as currently written would go away (though there may be mention of numbers of studies in the discussion) and the current introduction/discussion would need to be expanded and reorganized around common themes in the results from published papers. As I said before, there have been reviews that do this already for this topic, but they’re all a decade or so old by now, so there’s probably room for a new one – especially if you focus on what’s been added to the literature more recently.

Overall, while there is a lot of good information in this paper, its current format is somewhat confusing and unsatisfying.

---

## Round 0.2 · Major Revisions

· Academic Editor

Major Revisions

Please, consider the new comments of the reviewer 2. I do agree with that and certainly it is necessary addressing those issues. I think it would be useful to have a brief discussion at the end of the manuscript of what areas are most in need of more research. It could serve as a nice summary of where data is still needed in order to fully understand this topic.

Reviewer 1 ·

Basic reporting

This version of the ms includes most suggestion proposed, now it is adequate for publication.

Experimental design

This version of the review contains better curated information.

Validity of the findings

This version of the ms includes suggestion proposed; now it is sufficient for publication.

Additional comments

N/A

Reviewer 2 ·

Basic reporting

1. The introduction is long, a little jumbled, and not quite focused on the same thing as the rest of the paper, and as a result it feels disconnected from the rest of the paper. For example, there’s more than a full single spaced page of background on just herbivory in isolation, which seems like too much since that’s not the main point of this paper. I think the introduction needs to be made more concise, with relevant material shifted to the discussion section for more exploration and unnecessary material removed.
2. The introduction also probably needs to be restructured/reorganized. You want to give enough background information to directly lead into the topic of herbivory affecting plant fitness by affecting pollinator visitation, and lead up to your purpose in writing this review.
3. Related, there are a couple of places, especially in the introduction, that still cover several topics in one long paragraph, and probably need to be split up. For example, the second paragraph in the introduction (beginning at line 50) could be split into three paragraphs: one paragraph from line 50 to 56; a second paragraph from line 56 (starting with “Most typically…”) to line 67; and a third from line 67 (starting with “In this systematic review…”) to the end at line 72.
4. Compared to the first version of this paper, there have been significant improvements to content and organization of discussion section. I especially liked the first paragraph in the discussion. However, by adding so much material, it’s easy to get lost in all of the details when reading it. I suggest adding subheadings to split up the material (e.g. “Effects of florivory on pollination” and “Effects of vegetative herbivory on pollination” etc).
5. Line 354: “Folivory, root damage, and stem damage were found to negatively impact several floral traits, as well as pollinator visitation and reproduction.” You don’t really get into the details of what these effects are. I would prefer to see each of these types of vegetative damage talked about separately, with the kinds of effects on floral traits, pollination, and reproduction that are generally seen. If there is enough information, they may each get their own paragraph. If there is little information, they may just be separate sentences in the same paragraph.
6. Figure 3 is a nice addition. However, the red dots showing study sites are so small I didn’t see them at first even in color. In grayscale, they are completely gone.
7. I still have trouble with Figure 4. It’s very hard to read, with so many thing lines connecting names in small font, and having the lines behind the text makes the names harder to read. That said, it is improved from the last version in that I can at least (with some effort) tell which name each line is supposed to connect to.

Smaller comments:
• In the very first sentence in the introduction (line 50), “Plant fitness is determined by the net outcome of interactions with other species.” – while it’s true that interactions with other species partially determine plant fitness, they aren’t the only thing that affects plant fitness. Resource availability or abiotic factors, for example.
• Lines 44-46: “interactions between two species do not represent the net outcome of relationships” – these are typically called non-additive effects.
• Line 75: “rob the plant of resources” – the term “robbing” in the context of plant-insect interactions, especially involving flowers, may be confused for nectar robbing.
• Paragraph from line 73-85: in this paragraph introducing types of herbivory that can affect pollination, you talk about complete florivory, but not partial florivory – even though later you say partial florivory was more often studies.
• Line 169-171: You say that partial florivory can affect pollination by changing flower symmetry, but that’s far from the only effect of partial florivory on floral display. Published research has demonstrated partial florivory can affect flower size, color, VOCs, lifespan, sex ratio, nectar production, defenses in flowers, etc.
• Line 309: This is a comma splice at the comma between “potential to impact pollinators” and “for instance patches”
• Line 373: “directly comparing individual species” – species of what? Plants? Herbivores? Pollinators?
• Lines 438-439: This is a comma splice at the comma between “focal herbivores” and “those that looked”

Experimental design

My comments on the design of the previous version were mostly addressed. My only comments now are:
• You spend a lot of time talking about complete florivory versus partial florivory, but when you give stats (e.g. line 245, Figure 5) you don’t separate them out. This might be useful for comparison, especially since (as you point out) most florivory studies focus on partial florivory.
• I feel like I need to know a little more about your exclusion criteria, since there are a few citations I can think of that seem to measure direct effects of herbivory on pollination that aren’t included in your review.

Validity of the findings

• The part of your title after the colon (theories, methods, and challenges) doesn’t really reflect your findings or your purpose. You don’t cover current theories about resource allocation, pollinator selection criteria, or other theoretical frameworks that might have an impact on this topic. Similarly, you don’t really go over the methods for studies that test these kinds of questions. Challenges are only obliquely addressed, by pointing out what kind of studies haven’t yet been attempted much.
• Lines 298-301: You mentioned compensating for or shifting phenology in response to florivory, but there are also examples of species shifting mating system (i.e. from outcrossing to selfing in plants with mixed mating systems) that led to no overall reduction in reproductive output.
• Paragraph 363-383: Another potential explanation for the limited effect of vegetative damage on floral traits etc is that plants preferentially allocate resources to reproduction (since it’s so important), including flowers, and may suffer in other ways instead.
• You summarize your main findings at the end of the paper twice, in the paragraph beginning at line 448 and in the Conclusions section. Consider combining them into one conclusion section instead.

Additional comments

The authors’ revisions have, in my opinion, significantly improved this manuscript from the previous version. In particular, they added a lot more information overviewing previous findings on the topic in the discussion section, which lends itself much more to a review. However, despite significant edits, the introduction section still feels disconnected from the rest of the paper, and a little jumbled. The discussion would benefit from subheadings to provide structure. Revising the introduction may require major revision, but all of the other suggestions I’ve made are intermediate to minor revisions.

---

## Round 0.3 · accepted · Accept

· Academic Editor

Accept

After the second revision and the evident efforts made to include the comments and suggestions, particularly those from Reviewer 2, the manuscript is now ready for publication. The introduction has been substantially modified, and thus I congratulate the authors for a good job.